# Noise Disturbances and Calls for Police Service in València (Spain): A Logistic Model with Spatial and Temporal Effects

**DOI:** 10.3390/ijerph16162815

**Published:** 2019-08-07

**Authors:** Lia Seguí, Adina Iftimi, Álvaro Briz-Redón, Lucía Martínez-Garay, Francisco Montes

**Affiliations:** 1Department of Statistics and Operations Research, University of València, 46100 Burjassot, Spain; 2Health Services Research Unit, Foundation for the Promotion of Health and Biomedical Research of València Region (FISABIO), 46020 València, Spain; 3INCLIVA Health Research Institute, València, 46010 València, Spain; 4Statistics Office, City Council of València, 46002 València, Spain; 5Department of Criminal Law, University of València, 46022 València, Spain

**Keywords:** noise disturbances, resident complaints, logistic regression, spatio-temporal effects, socio-demographic and environmental effects, GIS

## Abstract

The purpose of this paper is to explore the presence of spatial and temporal effects on the calls for noise disturbance service reported to the Local Police of València (Spain) in the time period from 2014 to 2015, and investigate how some socio-demographic and environmental variables affect the noise phenomenon. The analysis is performed at the level of València’s boroughs. It has been carried out using a logistic model after dichotomization of the noise incidence variable. The spatial effects consider first- and second-order neighbors. The temporal effects are included in the model by means of one- and two-week temporal lags. Our model confirms the presence of strong spatio-temporal effects. We also find significant associations between noise incidence and specific age groups, socio-economic status, land uses, and recreational activities, among other variables. The results suggest that there is a problem of "social" noise in València that is not exclusively a consequence of coexistence between local residents. External factors such as the increasing number of people on the streets during weekend nights or during summer months severely increase the chances of expecting a noise incident.

## 1. Introduction

Sustained exposure to noise has been identified as a factor that can cause both auditory and non-auditory health effects [1]. Besides hearing loss, noise exposure can induce annoyance [2], cognitive impairment [3], sleep disturbance [4], cardiovascular diseases [5,6], and mental problems [7,8].

One of the challenges that modern European cities face is environmental noise, as has recently been pointed out by the World Health Organization Regional Office for Europe: “Noise is an important public health issue. It has negative impacts on human health and well-being and is a growing concern.” [9]. The first noise-abatement actions essentially consisted of legislation fixing maximum sound levels and technological progress: vehicles, airplanes, and machines had to comply with noise limits at the time of manufacture [10]. However, this policy could not solve completely the noise problem since the growth of road, rail, and air traffic were partly offsetting the technological improvements. This situation led to the enactment of the Directive 2002/49/EC of the European Parliament and of the Council whose aims were (1) “to define a common approach intended to avoid, prevent or reduce on a prioritized basis the harmful effects, including annoyance, due to exposure to environmental noise” and (2) to provide “a basis for developing Community measures to reduce noise emitted by the major sources, in particular road and rail vehicles and infrastructure, aircraft, outdoor and industrial equipment and mobile machinery”. In the case of València (Spain), the aforementioned Directive was transposed first into a regional law (*Ley 7/2002, de 3 de diciembre, de la Generalitat Valenciana, de Protección contra la Contaminación Acústica*) and then into a national legislation (*Ley 37/2003, de 17 de noviembre, del Ruido*).

Nevertheless, none of the instruments referred above targeted “social” noise (neighborhood noise) since it was thought to be dealt locally. This was the case of the City Council of València, which decided to issue an ordinance to solve the noise problem (*Ordenanza Municipal de Protección Contra la Contaminación Acústica* from June 26, 2008). This ordinance established, among other questions, the regulation of ZAS zones, i.e., areas in which the levels of noise exceed the limits allowed by the law. When an area is declared a ZAS zone (which literally means acoustically saturated zone, ZAS by its acronym in Spanish) the City Council can implement special measures in order to reduce environmental noise. These measures can be referred to the number of licensed premises (especially in the case of bars, pubs and nightclubs), the closing hours, the development of activities that may provoke noise, the limitation of the vehicle traffic or whatever measures considered appropriate.

Even though this ordinance was enacted in 2008, the problem of the noise has not been resolved yet. Evidence of this can be found in the fact that the Local Police of València records approximately 11,000 calls for service related to noise disturbances every year. As a result of the collaboration between the Local Police of València and the University of València in a larger project, a prior study of the calls for service for noise disturbances in València (Spain) was conducted [11]. The study covered a two-year period (2014–15) and the available data contained the following variables: time, date, and geographic location of the calls for service. A descriptive analysis showed that the calls were concentrated in specific boroughs and at specific times: mainly between 10 pm to 4 am during the weekends, with a peak in June. This pattern suggested that people’s recreational activities (and thus, “social” noise) could be the main cause of noise disturbances in València.

The current study aims to add to the knowledge of the “social” noise problem in two ways: (1) exploring the presence of spatial and temporal effects on the calls for service related to noise disturbances that were reported to the Local Police of València (Spain) in the period 2014–15 and (2) investigating the effect of socio-economic, demographic, and environmental characteristics of each borough. It is worth noting that many of the studies dealing with the investigation of the consequences of noise exposure for the population have employed objective measurements of noise levels. In particular, these studies have usually been focused on the noise generated by a predefined set of sources, commonly traffic-related. However, it is also common that studies of this kind lack data on the level of disturbance that such noise produces to residents. In this regard, it has been suggested that certain sounds or noise variations can be more disruptive and deleterious to cognitive development than being exposed to a constant noise level, regardless of the magnitude of it [12]. Therefore, the main novelty of our study in the context of analyzing the urban noise phenomenon is the entire focus we put on noise disturbance data generated by the own residents (representing what we have called “social” noise), rather than on global noise levels.

Regarding the choice of the method, a logistic model enables the estimation of the likelihood of a citizen calling the police to report a noise incident, but also permits to confirm the presence of both spatio-temporal and covariate effects on the existence of such calls. Thus, a logistic modelling approach allows constructing probabilistic maps over València representing the risk of noise disturbance, unveiling the factors that increase noise disturbance levels, and measuring the magnitude of the effect produced by each of those factors. All these outcomes provided by the logistic model can be helpful to establish preventive measures oriented towards reducing the magnitude of the noise problem in the city.

The choice of a logistic model to meet the research purposes we have established is well supported by multiple studies that were focused on crime data. For instance, some studies show the application of logistic regression models to predict the probability of burglary activities with respect to the event density epicenter, using a regular grid for event localization [13]. Other authors have investigated the potential of applying predictive analysis in an urban context through a logistic model and a neural network, using a raster grid [14]. Finally, logistic models have also been used to interpret insurgent attacks in Baghdad [15]. To test the hypothesis of heterogeneity, repeat victimization and denial policing the authors used a regular grid that decomposed the city of Baghdad into 3456 cells.

Unlike these three studies, our approach identifies the incidents over an irregular grid: the one formed by the boroughs of València. We chose this approach since all socio-economic and demographic variables for this administrative unit are available in the Statistics Yearbooks published by the Statistics Office of València [16].

## 2. Materials and Methods

### 2.1. Binomial Logistic Model

The occurrence of a noise incident can be described by a Bernoulli random variable, *Y*, such that Y=1 if an incident has occurred and Y=0 otherwise (Y∼B(1,p)). The effect of a set of variables, $\left\{X_{1},X_{2},\ldots ,X_{k}\right\}$, on *Y* can be modelled in this context by means of a *binomial logistic model*, in which the logarithm of the odds in favor of the noise call occurrence depends linearly on the variables. Namely,
(1)$$\mathrm{logit}\left(P\left(Y=1\right)\right)=\log\frac{P(Y=1)}{P(Y=0)}=\beta_{0}+\sum_{i=1}^{k}\beta_{i}x_{i},$$ where the variables can be numerical or categorical. If $X_{i}$ is categorical with *J* categories, the model includes it through J−1 dichotomous variables, $X_{ik},k=1,2,\ldots ,J-1$, so that $X_{ik}=1$ if $X_{i}=k$ and 0 otherwise. The missing category, which is taken as a reference and whose choice is arbitrary, is included in the model when $X_{ik}=0,\forall k$. This decomposition of $X_{i}$ avoids redundancy in the parameters.

Having fitted the model, P(Y=1) is estimated by means of $\pi_{1}$,
(2)$$\pi_{1}=\frac{\exp(\beta_{0}+\sum_{i=1}^{k}\beta_{i}x_{i})}{1-\exp(\beta_{0}+\sum_{i=1}^{k}\beta_{i}x_{i})}$$ an expression easily derived from (Equation 1).

### 2.2. Model Parameters Interpretation

The β coefficients in the model are directly related to the odds in favor of the occurrence of a call, which is simply called the odds ratio of the event of interest. In effect, if $X_{i}$ is a numerical variable, let us keep the rest of the variables constant and substitute in (Equation 1) the values $x_{i}$ and $x_{i}+1$, respectively. Taking antilogarithms and dividing both expressions, we will obtain,
$$\exp\left(\beta_{i}\right)=\frac{\frac{\pi_{1}\left(x_{i}+1\right)}{1-\pi_{1}\left(x_{i}+1\right)}}{\frac{\pi_{1}\left(x_{i}\right)}{1-\pi_{1}\left(x_{i}\right)}},$$ that is, $\exp(\beta_{i})$ is the change in the odds ratio when the variable increases one unit, the rest of the variables remaining constant.

If $X_{i}$ is a categorical variable with only two categories, represented by 0 and 1, from the quotient of the antilogarithms of (Equation 1) for both values of $X_{i}$, it follows that $\exp(\beta_{i})$ is the odds ratio of $X_{i}$ when the rest of the variables do not change.

Finally, if $X_{i}$ is a polytomous variable decomposed in dichotomous variables as above explained, $\exp(\beta_{ik})$ is the odds ratio for the category *k* and the reference category, as long as the rest of the variables do not change. An exhaustive presentation of logistic models can be found in [17].

### 2.3. Neighborhood Structure

A model that studies the spatial effect on the noise phenomenon, requires the definition of a suitable neighborhood structure for the spatial unit being used. Such a structure depends on the criteria used to define the concept of neighbor. If we define as *neighbors* those boroughs sharing a border, a criterion that seems the most appropriate for an irregular lattice such as the one we are considering, the neighborhood matrix *W* is defined by,
$$w_{ij}=\left\{\begin{array}{cc} 0, & i=j=1,\ldots ,n;\\ 1/n_{i}, & \mathrm{if}\;j\in V(i),\;\mathrm{with}\;n_{i}=\#V\left(i\right);\\ 0, & \mathrm{if}\;j\notin V(i), \end{array}\right.$$ where *i* and *j* represent two of the *n* boroughs and V(i) the set of neighbors of *i*. With this structure, no borough is its own neighbor, and the values in each row sum to unity because the weights $w_{ij}$ are standardized. For other neighborhood structures see [18].

### 2.4. Data

The 092 call center of the Local Police of València (PLV) receives complaints of incidents that occur in the city. During the years 2014 and 2015, around 480,000 incidents were recorded. For each of them time, date, type of incident, and location by geographic coordinates are known.

The aim of this work, as we have already pointed out in the Introduction, is the analysis of the incidents related to *noise* by using a logistic model that allows assessment of the presence of spatio-temporal effects and the influence of some variables on the phenomenon of noise. Therefore, the database is filtered to obtain only the information related to noise incidents.

A second filter is applied to exclude the incidents occurred in the boroughs belonging to districts 17 to 19 of the city of València, which are displayed in Figure 1 (left). These are three peripheral and sparsely populated districts which are located to the North, West, and South of the city, staying away from the urban core. Few calls from these districts are recorded, about 3.5% of the total of the two years analyzed. Hence, applying this second filter, a total of 22,419 calls are due to noise problems in districts 1 to 16, 11,577 in 2014 and 10,842 in 2015.

Figure 1 (right) shows the 70 boroughs of València in which districts 1 to 16 are divided and their identification numbers. A complete list of the names of all boroughs can be found in Table A1 in the Appendix A.

### 2.5. Data Transformation and Variables

The original database contains 22,419 calls related to noise incidents. Exact time of the event, day, and geographical location are recorded for each call. We divided the period of the day as follows: day (from 5 am to 21 pm) and night (from 22 pm to 4 am). To meet our objectives, this database is disaggregated on the basis of three dimensions: the period of the day on which the call can occur (day/night), the date (from 1 to 730 in order to cover the two years of study) and the borough where the call is located (70), which in turn gives rise to 102,200 (2·730·70) records. Thus, our dataset contains 102,200 records each of which incorporates the information shown in Table 1 for the corresponding combination of a period of the day, a date (from 2014 or 2015) and a borough.

We can distinguish in Table 1 three subsets of variables: those related to temporal effects (1 to 3), those that inform us about incidents that occurred in previous weeks in the borough and its neighbors (4 to 9) and those describing the characteristics and environment of the borough, which include the district within which the borough is placed (10), demography (11 to 15), services that may represent noise sources (16), a socio-economic index (17), traffic-related information (18), age of the buildings (19), land uses (20 to 21) and presence of “botellón” frequent locations within borough boundaries (22). All the variables corresponding to borough characteristics (11 to 22) vary yearly (for 2014 or 2015).

The district to which each borough belongs to is used as a categorical variable in order to capture a part of the spatial heterogeneity of the noise phenomenon in València that may be missed by the variables 11 to 22 (which are all obtained at the borough level). The choice of the district instead of the borough itself is due to avoid model overfitting issues.

The socio-economic vulnerability index is a continuous variable that takes values in the interval [1,5] and the lower its value, the more vulnerable the borough. It is a synthetic index made up from 10 variables which considers, among others, the academic level of the residents, the value of vehicles and dwellings registered/located in the borough and an approximation of average income at the district level. Details can be found in [19].

“Botellón” is a popular phenomenon among Spanish teenagers and young adults that consists of gathering in public spaces to socialize and drink alcohol [20,21]. The binary variable related to “botellón” was constructed and validated following several media and police sources, being highly correlated in space with the presence of pubs and nightclubs across the city.

Finally, variable number 23 in Table 1 is the number of noise calls that take place in the borough, which is converted into a binary variable to become the response of the logistic model. Hence, in view of the variables indicated in Table 1, the logistic model chosen for this research follows the next mathematical expression,
(3)$$\begin{array}{ccc} \mathrm{logit}(\mathrm{noisebin}) & = & \beta_{0}+\beta_{1}\mathrm{weekday}+\beta_{2}\mathrm{month}+\beta_{3}\mathrm{period}+\beta_{4}{\mathrm{noisebin}}_{1}+\beta_{5}{\mathrm{noisebin}}_{2}\\ & & +\beta_{6}{\mathrm{noisebin}}_{11}+\beta_{7}{\mathrm{noisebin}}_{12}+\beta_{8}{\mathrm{noisebin}}_{21}+\beta_{9}{\mathrm{noisebin}}_{22}+\beta_{10}\mathrm{district}\\ & & +\sum_{k=11}^{22}\beta_{k}{\mathrm{variable}}_{k}. \end{array}$$

The response variable, *noisebin*, and the rest of variables related to noise in (Equation 3), *noisebin*${}_{t}$, and *noisebin*${}_{ts}$ with t=1,2 and s=1,2, are categorical binary variables derived from the original noise variables defined in Table 1. These variables take the value 0 if the corresponding *noise* variable is 0, and 1 otherwise. The subscripts *t* and *s* stand for the temporal and spatial lag, respectively.

## 3. Results

This section presents the outcomes and interpretations that derive from the use of the statistical model represented by Equation (Equation 3), based on the following three hypotheses regarding the occurrence of noise disturbance events,
the presence of spatial and temporal (*weekday*, *month* and *period*) effects,the effect that incidents happened one or two weeks earlier in the borough neighborhoods, *noise*${}_{ts}$, and in the borough itself, *noise*${}_{t}$, have on what occurs now in the borough, andthe effect that a set of variables linked to the borough has on the occurrence of noise incidents.

The model has been fitted using the *glm* function of the stats package of statistical software R [22]. Table 2 analyzes the significance of changes in model deviance as variables are being added. The column *Deviance* shows the deviance reduction when adding each variable to the model. The significance of this reduction is contrasted by means of a $\chi^{2}$-test with the degrees of freedom associated with the variable (column *Df*). Despite having been included in the final model, *noisebin${}_{12}$* does not reduce model deviance significantly.

Table 3 shows the result of the fit, with the odds ratio associated with each coefficient, exp(β), and its 95% confidence interval (CI). *Sunday*, *June* and *District 1* appear in blank in the table because they are the categories taken as reference for the variables *weekday*, *month* and *district*, respectively.

First, note in Table 3 that the coefficients of the levels associated with weekdays, months, and districts are all negative. This is because the reference categories are those with the highest number of incidents [11]. A negative coefficient means that odds favorable to the existence of calls due to noise incidents are lower in those days, months, or districts than those in the reference category. Regarding the variable representing the two periods considered for each day, the odds of the night are 88% higher than those of the day, as $\exp(\beta_{\mathrm{night}})=1.88$.

The statistical significance of these findings allows validation of the first of the hypotheses that we had established: there are spatial and temporal effects that drive noise disturbance events in València. Furthermore, to better appreciate the influence of space and time in the occurrence of noise incidents, the model can be used to map noise call probabilities at the borough level. Figure 2 shows two very disparate scenarios in relation to the noise phenomenon: the probabilities estimated for Monday nights in the month of January and those estimated for Sunday nights in June (considering the year 2015). According to these two maps, the existence of both spatial and temporal effects seems clear. Indeed, most of the boroughs presenting the highest probabilities are in the city center of València (District 1) or in two of its neighboring districts (Districts 2 and 3). The temporal effects are even more evident: only the probability of Borough 112 exceeds 0.3 for Monday nights of January, whereas a total of 20 boroughs get an estimation over 0.5 for Sunday nights within the month of June.

The sign and significance of the coefficients of the variables that represent the occurrence of noise incidents in previous weeks in the borough and its neighborhood indicate the existence of a positive spatio-temporal effect, i.e., the occurrence of previous incidents makes more likely that they occur again. The effects of the temporal lag of order 2 considering the borough and its first-order neighborhoods, *noisebin*${}_{2}$ and *noisebin*${}_{21}$, are the most pronounced, with increments of 22% and 20% in the corresponding odds ratios. On the other hand, the variable *noisebin*${}_{12}$ is the only one from this group lacking statistical significance, which is not surprising in view of Table 2.

It is important to note that the inclusion of lagged variables representing the history of the phenomenon of interest in close space and time is vital to produce variability in model predictions. Indeed, in the absence of lagged variables, the sequence of probabilities predicted for the whole set of boroughs under any scenario would always keep the same order relationships. As an illustration, the boroughs highlighted in each of the maps shown in Figure 2 are the five presenting the highest probabilities for the occurrence of a noise call event. Although both subsets share three boroughs (16, 21 and 112), the use of the lagged variables is what generates variations in the other two members.

Most of the variables linked to the demographic composition of the neighborhood, log(*inhab*), *inhab14*, *inhab1529* and *inhab65*, show significant effects and a behavior consistent with what could be expected, positive coefficients. Regarding the interpretation of their odds ratios, one needs to account for the magnitude of each variable (a caution that was not required when considering the variables discussed in the former paragraphs because all of them were categorical). For instance, the variable log(*inhabitants*) presents a very high odds ratio of 2.52. Hence, a unit increase in log(inhabitants), which implies an unrealistic population growth factor of e≈2.7183, will lead to a 152% increase in the odds ratio. A more probable increase in the population of, say, the 10%, produces an odds ratio of only 1.09.

Other characteristics of the boroughs also produce significant and reasonable parameter estimations. The variables *barrest*, *educuse* and *botellón* display a positive association with odds ratio. In particular, the contribution of “botellón” to the noise problem is remarkable.

On the other side, the socio-economic vulnerability index and the percentage of land that is used for green areas present a negative coefficient that indicates a reduction in the odds ratio of noise calls. The variable *mainroad* receives a positive but non-significant estimation.

Finally, the variables *mphouse* and *buildage* display a more controversial result. The demographic variable *mphouse* shows a significant and negative association with odds ratios, even though it is positively correlated with *inhab14* and *inhab1529* (Figure 3). Data exploration helps to check that some of the highest values of *mphouse* take place at peripheral and calm boroughs of the city, which may be confounding the interpretation of this variable. Regarding *buildage*, one may expect a positive association with the noise phenomenon, as older buildings should be, on average, less acoustically isolated. Moreover, old buildings are more concentrated around the city center of the city, which appears to be the zone that is most affected by the noise phenomenon. Therefore, we find difficult to explain the effect detected for this variable.

## 4. Discussion

The results confirm the main hypotheses set out in the research: the existence of spatio-temporal effects on noise calls and the influence that certain variables representing some characteristics of the boroughs of València have on the phenomenon. We now discuss several aspects related to both dimensions of the problem in more detail. Before further discussing the results, it is worth remembering that the data we have analyzed originated from calls made to the Local Police of València by the residents of the city to complain of a noise problem. Thus, what a resident perceives as disturbing noise may not correspond to exceeding the thresholds allowed by the legislation regulating noise levels. This clarification is relevant as it can be helpful to better explain or understand some of the results provided by the logistic model.

Regarding spatio-temporal effects, the model applied shows that what happened in a borough or its neighbors in the previous two weeks positively influences the existence of new calls. We already highlighted that this fact is key to allow variability in model predictions, keeping the model updated according to recent calls. Furthermore, it is clear that late-night calls are more likely, and that noise-related calls are more likely to occur as the week goes by, with the weekend as the highest point. The months of May to September, with June leading the way, are also more likely to register noise-related disturbances than the rest of the year. What the model seems to be pointing out, albeit indirectly, is that the problems of noise in a borough are not exclusively problems of coexistence, and therefore generated by its residents, but that there is an external factor due to the greater presence of people on the streets on weekend nights, which increases during the summer months. València is a city located on the shores of the Mediterranean, approximately 39${}^{\circ }$ North latitude, with a mild climate. The noise problem is greater in those areas of the city with greatest number of leisure venues where people from different parts of the city gather. This conclusion is reinforced by our model as the covariate describing the density of bars and restaurants presents a significant positive effect.

Concerning the factors that may generate noise annoyances among the residents of an urban area, vehicle traffic is largely known as one of the most important [24,25,26]. The only traffic-related variable included in our model did not receive a significant estimation. At this point, it is of need to remark again that most of the research on the noise phenomenon relies on physical measures as the equivalent continuous sound level (Leq), measured in decibels (dB). There should be an association between “real” noise levels at a borough over a certain period of time and the number of calls that people make [27], but it is obvious that both approaches to the noise problem could lead to different conclusions. In particular, it is not clear, at all that the boroughs of València that suffer from heavy traffic are generating many police calls on this issue. Taking this into consideration, we discuss some other interesting findings regarding borough characteristics and noise disturbance calls.

Our model also confirms the existence of socio-demographic effects on the calls for service for noise disturbances. It stands to reason that an increase in the number of inhabitants can affect the number of calls for service. With respect to the variables representing some population age groups, it is plausible that youngest and oldest cohorts are particularly implicated in noise calls: the first, as likely noise generators, the second, as likely callers because they are usually more sensitive to noise.

Socio-economic status, measured through a socio-economic vulnerability index, shows a negative association with noise disturbances. This result is consistent with some previous research works that have found higher levels of noise exposure at disadvantaged areas [28], but it differs from other research outcomes also developed in the context of a Spanish city [26].

Land uses are usually employed for modelling noise levels [29]. We considered educational and green land uses. The first shows a positive relationship with the odds of a noise call, which seems reasonable as children and young adults (university students) tend to meet in the surroundings of the buildings where they share their studies. Green areas display a negative association with noise events, which agrees with research suggesting that green spaces are capable of mitigating noise levels [30].

In conclusion, the choice of a logistic modelling approach has enabled us to verify the spatio-temporal nature of the noise disturbance phenomenon in València, and to identify certain social and environmental factors that are associated with higher levels of noise disturbance. From these results, and on the basis of the density pattern of the noise calls occurred in València shown in [11], we can assure that noise disturbance events follow a non-random spatial distribution across the city. The existence of clusters is also obvious, as well as certain rejection dynamics. All these facts suggest future lines of work in the field of spatio-temporal point processes that may be helpful to provide a deeper knowledge of the noise problem. A log-Gaussian Cox model such as the one described in [31] or a multi-scale area-interaction model [32] may be used to model the point pattern of the noise phenomenon.

Furthermore, the elaboration of probabilistic maps of the risk of noise disturbance as those shown in the previous section (which make use of updated information on the problem through spatial and temporal lags), can help authorities to anticipate the presence of some “social” noise generator in certain boroughs of the city and act accordingly, providing a tool to the police for reducing the magnitude of the problem in the short term. In the long term, public policies should be oriented towards diminishing noise levels by better accounting for some of the factors that have been associated with increased odds of producing a noise call in València’s boroughs. A major control of the licensing of certain services, such as bars and restaurants, putting more effort into civic education (to reduce the “botellón” phenomenon) and to promote the creation of green areas that may act as natural noise mitigators (besides producing other health benefits), are some of the strategies that could be attempted for getting closer to the solution of the noise problem currently existing in València.

## Figures and Tables

**Figure 1 ijerph-16-02815-f001:**
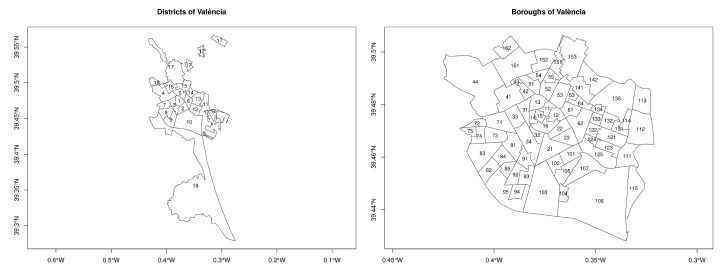
Districts (**left**) and boroughs (**right**) of València.

**Figure 2 ijerph-16-02815-f002:**
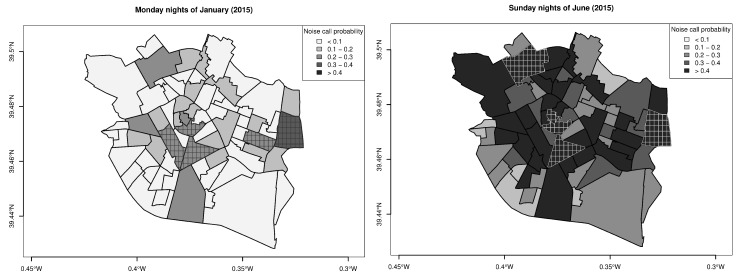
Probability of a noise call estimated for Monday nights of January (**left**) and Sunday nights of June (**right**) in the boroughs of València (year 2015). The five boroughs with the highest probabilities under each of the two scenarios are highlighted with a striped pattern.

**Figure 3 ijerph-16-02815-f003:**
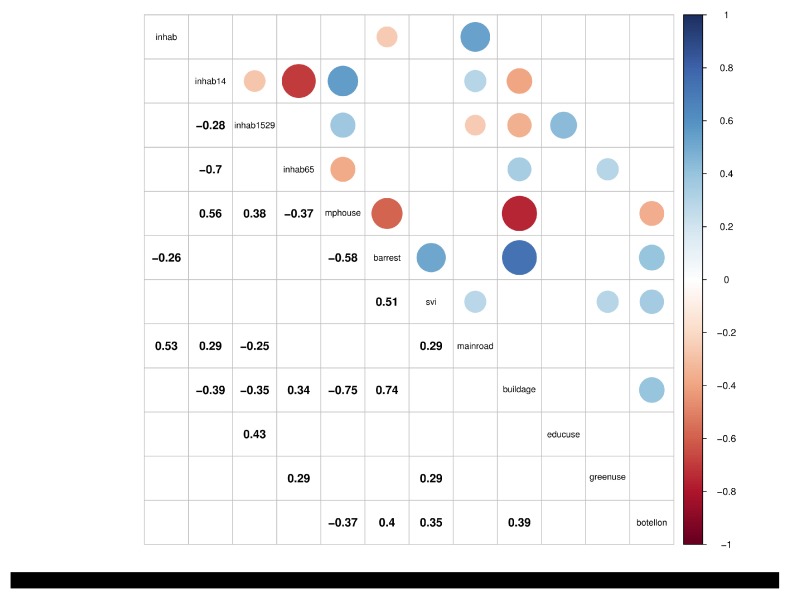
Correlation matrix for the variables representing borough characteristics. Only the correlation coefficients that are significant at the 0.05 level are shown. Figure generated with the R package corrplot [23].

**Table 1 ijerph-16-02815-t001:** Enumerated list of variables used in the analysis and their meaning.

Variable	Meaning
(1) weekday	day of the week
(2) month	month of the year
(3) period	period of the day
(4) noise${}_{1}$	number of noise calls 1 week before in the borough
(5) noise${}_{2}$	number of noise calls 2 weeks before in the borough
(6) noise${}_{11}$	number of noise calls 1 week before in the 1 lag borough neighborhood
(7) noise${}_{12}$	number of noise calls 1 week before in the 2 lag borough neighborhood
(8) noise${}_{21}$	number of noise calls 2 week before in the 1 lag borough neighborhood
(9) noise${}_{22}$	number of noise calls 2 week before in the 2 lag borough neighborhood
(10) district	district to which the borough belongs to
(11) inhab	number of inhabitants in the borough
(12) inhab14	percentage of inhabitants in the borough aged under 15
(13) inhab1529	percentage of inhabitants in the borough aged between 15 and 29
(14) inhab65	percentage of inhabitants in the borough aged 65 or over
(15) mphouse	average number of members per household
(16) barrest	number of bars and restaurants per 100 inhabitants in the borough
(17) svi	socio-economic vulnerability index
(18) mainroad	number of km of non-pedestrian main road located in the borough
(19) buildage	average age of the buildings located in the borough
(20) educuse	percentage of land in the borough dedicated to educational use (undergraduate and university level)
(21) greenuse	percentage of land in the borough dedicated to green areas
(22) botellón	binary variable indicating if the practice of “botellón” is usual in the borough
(23) noise	number of noise calls in the borough

**Table 2 ijerph-16-02815-t002:** Model deviance analysis.

	Df	Deviance	Resid Df	Resid Dev	*p*-Value
NULL			100239	88,454.79	
weekday	6	1692.29	100233	86,762.50	0.00
month	11	918.26	100222	85,844.23	0.00
period	1	1409.47	100221	84,434.76	0.00
noisebin${}_{1}$	1	319.50	100220	84,115.26	0.00
noisebin${}_{2}$	1	361.51	100219	83,753.75	0.00
noisebin${}_{11}$	1	255.37	100218	83,498.38	0.00
noisebin${}_{12}$	1	1.85	100217	83,496.53	0.17
noisebin${}_{21}$	1	393.44	100216	83,103.09	0.00
noisebin${}_{22}$	1	18.69	100215	83,084.39	0.00
district	15	296.35	100188	77,919.11	0.00
log(inhab)	1	2320.00	100214	80,764.40	0.00
inhab14	1	180.97	100213	80,583.42	0.00
inhab1529	1	381.78	100212	80,201.65	0.00
inhab65	1	29.17	100211	80,172.47	0.00
mphouse	1	1111.22	100210	79,061.26	0.00
barrest	1	365.69	100209	78,695.57	0.00
svi	1	167.76	100208	78,527.81	0.00
mainroad	1	43.18	100207	78,484.63	0.00
buildage	1	60.84	100206	78,423.79	0.00
educuse	1	10.04	100205	78,413.74	0.00
greenuse	1	109.99	100204	78,303.75	0.00
botellón	1	88.30	100203	78,215.46	0.00

**Table 3 ijerph-16-02815-t003:** Results of the logistic model.

Variable	β	SE	*p*-Value	exp(β)	95% CI exp(β)	
					Lower	Upper
Monday	−0.89	0.04	0.00	0.41	0.38	0.44
Tuesday	−0.83	0.04	0.00	0.43	0.40	0.46
Wednesday	−0.76	0.03	0.00	0.47	0.43	0.50
Thursday	−0.70	0.03	0.00	0.50	0.46	0.53
Friday	−0.45	0.03	0.00	0.64	0.60	0.68
Saturday	−0.08	0.03	0.01	0.92	0.87	0.98
Sunday						
January	−1.00	0.05	0.00	0.37	0.33	0.40
February	−0.83	0.05	0.00	0.43	0.40	0.47
March	−0.55	0.04	0.00	0.58	0.53	0.62
April	−0.65	0.04	0.00	0.52	0.48	0.57
May	−0.44	0.04	0.00	0.65	0.59	0.70
June						
July	−0.28	0.04	0.00	0.75	0.70	0.81
August	−0.48	0.04	0.00	0.62	0.57	0.67
September	−0.35	0.04	0.00	0.70	0.65	0.76
October	−0.49	0.04	0.00	0.61	0.56	0.66
November	−0.74	0.04	0.00	0.47	0.43	0.52
December	−0.78	0.04	0.00	0.46	0.42	0.50
night	0.63	0.02	0.00	1.88	1.81	1.96
noisebin${}_{1}$	0.13	0.02	0.00	1.14	1.08	1.19
noisebin${}_{2}$	0.20	0.02	0.00	1.22	1.17	1.28
noisebin${}_{11}$	0.07	0.02	0.00	1.08	1.03	1.12
noisebin${}_{12}$	−0.03	0.03	0.40	0.97	0.90	1.04
noisebin${}_{21}$	0.18	0.02	0.00	1.20	1.14	1.25
noisebin${}_{22}$	0.10	0.04	0.00	1.11	1.03	1.19
log(inhab)	0.93	0.03	0.00	2.52	2.39	2.65
inhab14	0.13	0.01	0.00	1.14	1.11	1.17
inhab1529	0.12	0.01	0.00	1.13	1.10	1.16
inhab65	0.04	0.01	0.00	1.04	1.03	1.05
mphouse	−2.04	0.16	0.00	0.13	0.09	0.17
barrest	0.18	0.02	0.00	1.20	1.15	1.24
svi	−0.38	0.03	0.00	0.68	0.64	0.73
mainroad	0.02	0.01	0.06	1.02	1.00	1.03
buildage	−0.00	0.00	0.00	1.00	0.99	1.00
educuse	0.01	0.00	0.00	1.01	1.00	1.01
greenuse	−0.02	0.00	0.00	0.98	0.98	0.99
botellón	0.29	0.03	0.00	1.33	1.25	1.42
District 1						
District 2	−0.76	0.07	0.00	0.47	0.40	0.53
District 3	−1.02	0.07	0.00	0.36	0.31	0.42
District 4	−1.05	0.10	0.00	0.35	0.28	0.42
District 5	−1.07	0.10	0.00	0.34	0.28	0.41
District 6	−0.81	0.10	0.00	0.44	0.35	0.54
District 7	−1.23	0.11	0.00	0.29	0.23	0.35
District 8	−1.22	0.11	0.00	0.30	0.23	0.36
District 9	−1.38	0.11	0.00	0.25	0.20	0.30
District 10	−1.10	0.10	0.00	0.33	0.27	0.40
District 11	−1.29	0.11	0.00	0.28	0.22	0.34
District 12	−1.04	0.10	0.00	0.35	0.28	0.43
District 13	−1.13	0.11	0.00	0.32	0.25	0.39
District 14	−1.47	0.12	0.00	0.23	0.18	0.29
District 15	−1.37	0.13	0.00	0.25	0.19	0.32
District 16	−1.23	0.12	0.00	0.29	0.22	0.36

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
