# Peer review of "Noise Disturbances and Calls for Police Service in València (Spain): A Logistic Model with Spatial and Temporal Effects"

_ijerph, 2019, doi:10.3390/ijerph16162815_

Round 1
Reviewer 1 Report
The thesis is original, but it must be explained better. There is a big discussion, but a clear conclusion is missing. There is a big chapter of materials and
methods with good technical observations, but it is not clear the correlation with results. The interpretation of
data is deep, but the reader doesn't understand very well what you want
to express. Explain better the sociological purposes of the scientific paper.
Author Response
We have incorporated several paragraphs into our manuscript to try to address your concerns. In the following list we highlight all the modifications carried out.
· The introduction of the manuscript has been expanded. We have included in the new version several references to some of the health problems that can arise as a consequence of a sustained exposure to noise.
· In the introduction we have remarked what we think is the main novelty of our study, in comparison to other studies on the noise phenomenon: we have analyzed data generated by the citizens of València that have suffered from noise disturbance during a period of two years, instead of global noise measurements (which is the most usual approach in similar research studies).
· We have better highlighted the advantages of using the logistic modelling approach. In particular, the logistic model allows the construction of probabilistic maps that show how the risk of noise calls distributes across the city. This can be used by local authorities to implement preventive measures.
· In the conclusion we have indicated how our research could be helpful to face the noise issue currently existing in València. Probabilistic maps that make use of updated information of the noise calls received (which are considered in the logistic model through spatial and temporal lags) can be used by Police officers to reduce the magnitude of the problem in the short term, by acting in the boroughs showing a higher probability of a noise call. In the long term, public policies should account for several of the factors that we have identified as the ones that increase the likelihood of a noise call at the borough level.
Thank you very much for your useful suggestions.
Reviewer 2 Report
The presented article concerns a very important issue. However, the scale of the problem and a huge number of publications related to issues raised in the article, requires from the Authors the particular accuracy and precision in presentation of the research results.
In my view, therefore, before any publication the attempt to sort out the text should be taken once again. First of all it is necessary to define the purpose to determine the fundamental issue as well as to consider how the described research increased the knowledge about the problem.
The reviewed article, in my opinion, is missing also reliable conclusions, which may be the consequence of a little emphasized aim of this publication. The type of publication accepted by Authors - case study - is a being the analysis of the particular case publication, giving the possibility to draw conclusions about the causes and results of the case described in it- the description of the case before 2014 and after 2015.
Author Response

(The authors gave the same response as above.)

Reviewer 3 Report
I applaud this undertaking in that it was a joint effort by researchers, the police department and the 092 Call Center to better identify where and when noise calls are made. The methodology employed, the way the data were presented, the interpretation of the results and the discussion that followed were readily understood. However, there has to be greater purpose to why one would undertake a study that would cast light on a problem, in this case noise pollution, in the city of Valencia. Thus,I would like to see the authors suggest ways their study could be used to address the noise problem in Valencia with the goal of lessening noise pollution. Recommendations are called for and this should include how the Police Department and Call Center could contribute in reducing the noise in the city. The authors in their introduction have failed to discuss the deleterious impacts of noise on health. Such an introduction would strengthen the need for the research they conducted. I would suggest adding adverse noise impacts to the introduction. I do not consider these suggestions "major" in that they do not speak to the manner in which the research was conducted. If anything, in my opinion, following these suggestions will strengthen the value of this study.Author Response
We have incorporated several paragraphs into our manuscript to try to address your concerns. In the following list we highlight all the modifications carried out.
· The introduction of the manuscript has been expanded. We have included in the new version several references to some of the health problems that can arise as a consequence of a sustained exposure to noise.
· In the introduction we have remarked what we think is the main novelty of our study, in comparison to other studies on the noise phenomenon: we have analyzed data generated by the citizens of València that have suffered from noise disturbance during a period of two years, instead of global noise measurements (which is the most usual approach in similar research studies).
· We have better highlighted the advantages of using the logistic modelling approach. In particular, the logistic model allows the construction of probabilistic maps that show how the risk of noise calls distributes across the city. This can be used by local authorities to implement preventive measures.
· In the conclusion we have indicated how our research could be helpful to face the noise issue currently existing in València. Probabilistic maps that make use of updated information of the noise calls received (which are considered in the logistic model through spatial and temporal lags) can be used by Police officers to reduce the magnitude of the problem in the short term, by acting in the boroughs showing a higher probability of a noise call. In the long term, public policies should account for several of the factors that we have identified as the ones that increase the likelihood of a noise call at the borough level.
Thank you very much for your useful suggestions.
Round 2
Reviewer 1 Report
Now the paper is really good.